# Climate Variations in the Low-Latitude Plateau Contribute to Different Sugarcane (*Saccharum* spp.) Yields and Sugar Contents in China

**DOI:** 10.3390/plants12142712

**Published:** 2023-07-21

**Authors:** Yong Zhao, Ling-Xiang Yu, Jing Ai, Zhong-Fu Zhang, Jun Deng, Yue-Bin Zhang

**Affiliations:** 1Sugarcane Research Institute, Yunnan Academy of Agricultural Sciences, Kaiyuan 661699, China; 18087395132@163.com (Y.Z.); zhongfu_zhang@yeah.net (Z.-F.Z.);; 2National Key Laboratory for Biological Breeding of Tropical Crops, Kunming 650205, China; 3Climate Center, Yunnan Meteorological Bureau, Kunming 650034, China; yulx@163.com

**Keywords:** sugarcane (*Saccharum* spp.), climate condition, average air temperature, average relative humidity, average rainfall amount, average sunshine duration

## Abstract

In China, the main sugarcane (*Saccharum* spp.) planting areas can be found in the low-latitude plateau (21° N–25° N, 97° E–106° E), which has most of the natural ecological types. However, there is limited information on the climate conditions of this region and their influence on sugarcane yield and sucrose content. Monthly variations in the main climate factors, namely, average air temperature (AAT), average relative humidity (ARH), average rainfall amount (ARA), and average sunshine duration (ASD), from 2000 to 2019 and sugarcane yield and sucrose content of 26 major sugarcane-producing areas from 2001/2002 to 2018/2019 were collected from the low-latitude plateau in Yunnan for studying the impact of climate variations on sugarcane yield and sucrose content. The results showed that AAT in the mid-growth season had a significant positive correlation with sucrose content (*p* < 0.05), and AAT in the late-growth season had a very significant positive correlation with sucrose content (*p* < 0.01). ARH in the mid-growth season had a significant positive correlation with sugarcane yield (*p* < 0.05). ARA in the early-growth season showed a significant positive correlation with sugarcane yield (*p* < 0.05). ASD in the late-growth season had a significant positive correlation with sugarcane yield (*p* < 0.05) and sucrose content (*p* < 0.01). The rainy and humid sugarcane areas were characterized by high ARA and ARH during the entire growth period, low AAT and ASD in the mid-growth season, and low AAT in the late-growth season, contributing to a high sugarcane yield, but not a high sucrose content. The low temperature and sunshine semi-humid sugarcane areas were characterized by the lowest AAT in the early and middle stages of sugarcane growth, less ASD in the early and middle stages, and less ARA in the early and late stages, which are unfavorable for sugarcane yield and sucrose content. The high temperature and humidity sugarcane areas were characterized by higher AAT and ARA, and moderate ASD during the entire growth period, resulting in good sugarcane growth potential and contributing to the sugarcane yield and sucrose content. The semi-humid and multi-sunshine sugarcane areas were characterized by the lowest ARH in the entire growth period, the lowest ARA in the middle and late seasons, and the longest ASD, contributing to an increase in sucrose content. The humid and sunny areas were characterized by the longest ASD and high ARH in the early and late seasons of sugarcane growth and moderate AAT and ARA during the entire growth season, which are beneficial for high sugarcane yield and sucrose content. Overall, these findings suggest that the sugarcane variety layout should be based on the climate type (of which there are five in the plateau), and corresponding cultivation practices should be used to compensate for the climatic conditions in various growth stages.

## 1. Introduction

Sugarcane (*Saccharum* spp.) is a perennial tropical and subtropical herb and a C4 crop, and it is an important raw material for sugar production worldwide. Because of its strong photosynthetic capacity, high biological yield, and total fermentable sugar, sugarcane has shown good developmental prospects for utilization as an energy material [1]. Currently, sugarcane is mainly used for sugar production in China, accounting for 90% of the total sugar yield. Increasing the sugar yield has always been an important aim of sugarcane planting; this is mainly affected by the biomass yield and sugar content of sugarcane, which depend on many factors such as climate and meteorological factors [2], varieties [3], cultivation measures [4], soil fertility [5], and pest control [6]. Meteorological factors directly affect growth and regional distribution of sugarcane planting [7]. The climate of Yunnan basically belongs to the subtropical plateau monsoon type, with remarkable three-dimensional climate characteristics. There are many climate types in Yunnan, with small annual temperature differences, large daily temperature differences, distinct dry and wet seasons, and obvious vertical variations in temperature with terrain height. Yunnan has cold, warm, and hot (including the subtropical zone) climates. In Yunnan, the low-latitude plateau (21° N–25° N, 97° E–106° E) is one of the most important sugarcane-planting areas and has most of the ecological types in China [8]. The sugarcane areas show diverse climate types, large climate differences, and significant differences in sugarcane yield and sucrose content. A comprehensive understanding of the climate characteristics of this region can provide basic data support for improving sugarcane cultivation measures, implementing an optimized variety layout, increasing the sucrose yield, and tapping the sugarcane-production potential. 

Climate conditions have a significant impact on sugarcane yield and sucrose content, of which temperature, rainfall, humidity, and sunshine are important factors that limit sugarcane production [9]. Previous studies have shown that sunshine and accumulated temperature have a very significant positive correlation with sugarcane yield [10], and rainfall has a key impact on sugarcane elongation and maturity, and a positive correlation with yield [9]. Aboveground dry weight, cane yield, and cane sucrose yield are significantly related to temperature and rainfall [11]. Atmospheric relative humidity, temperature, and sunshine are important factors that affect sugarcane elongation, sucrose accumulation, and yield [12]. Fabio and Gustavo used multi-year data from 37 meteorological stations to simulate and evaluate the potential and achievable yield of sugarcane and found that climate factors could explain 43% variability in sugarcane yield efficiency [13]. The order of importance of meteorological factors that affect yield is solar radiation, water shortage, maximum temperature, rainfall, and minimum temperature. Araujo et al. found that rainfall was an important factor that affected the yield of sugarcane [14]. The sucrose content of sugarcane is more vulnerable than the yield to the impact of air temperature; the process of sucrose accumulation in stems is affected by many factors, especially climate conditions such as temperature and water [15]. 

Sugarcane has a prolonged growth cycle that is typically divided into five stages: germination, seedling, tillering, elongation, and maturity. The germination and seedling stages are the early stages of sugarcane production, which are characterized by low temperature and inadequate rainfall in the regions of the low-latitude plateau. The middle stage of sugarcane growth comprises tillering and elongation, and it is characterized by elevated temperature and abundant rainfall. The maturity stage is the final phase of sugarcane development; this period is crucial for sugarcane yield and sucrose content, with a large difference in temperature between morning and evening, and moderate rainfall. The growth characteristics of sugarcane are variable across different time periods and are significantly influenced by climatic factors [8]. Previous studies [12] have extensively investigated the impact of climatic factors on sugarcane production. These studies used various methodologies, including relevant models, to assess sugarcane production [10], conducted regression analyses to establish a correlation between climatic factors and sugarcane yield [11], and examined the impacts of climatic factors during specific planting periods on sugarcane growth [14].

However, few studies have examined the impacts of climatic factors on the overall yield and sucrose content of sugarcane at different growth stages on a large scale. Additionally, there is a lack of information on the distribution of sugarcane yield and sucrose content potential in relation to specific climatic characteristics. Moreover, the ecological characteristics of sugarcane-growing areas in the low-latitude plateau are complex, and the climate characteristics of each sugarcane-growing area are unique. There is limited comprehensive data regarding changes in the characteristics of the main climate-influencing factors. Furthermore, few studies have elucidated the impacts of climate change on sugarcane yield and sugar content in each growth period. Therefore, the appropriate distribution of sugarcane varieties and use of cultivation measures in different growing regions need to be considered for improving both yield and sucrose content, particularly in the context of climate change. The objectives of this study were to (1) understand the detailed characteristics of the ecological climate in the low-latitude plateau sugarcane region, (2) comprehend the characteristics of the main climate factors in different periods of sugarcane growth and their correlation with sugarcane yield and sucrose content, and (3) delicately divide the ecological types of sugarcane areas and understand the impact of current climate conditions on sugarcane production in the low-latitude plateau.

## 2. Results

### 2.1. Differences in Climate Factors at Different Sites

From 2000 to 2009, monthly AAT, ARH, ARA, and ARD values of the test sites in 26 counties in seven prefectures in the low-latitude plateau of Yunnan (classified and analyzed by administrative regions) were statistically analyzed (Figure 1). Differences were detected in monthly AAT, ARH, ARA, and ARD of 26 test sites, but the overall trend was consistent. Monthly AAT was the lowest in December and January (5–15 °C), highest in June, July, and August (20–25 °C), increased month by month from February to June, and declined significantly from August to January. Monthly ARH was the highest in June, July, and August (75% to 90%), lowest in March (50% to 70%), increased month by month from March to July, and decreased month by month from July to February. For monthly ARA, obvious differences in monthly ARA values were observed among the main producing counties. Generally, monthly ARA was the highest in July (between 150 mm and 500 mm), lowest in November to March (between 0 mm and 50 mm), mainly concentrated in May to October, increased month by month from March to July, and decreased month by month from July to February. For monthly ASD, obvious differences in monthly ASD values were observed among the test sites. Monthly ASD was the shortest in July (60–150 h), longest from November to March and from January to March (100–260 h), decreased month by month from March to July, and increased month by month from July to February. Monthly ARH, ARA, and ARD showed a consistent trend among the test sites (counties) in the same prefecture, but differed in different prefectures. Monthly AAT, ARH, ARA, and ARD of 26 test sites were obviously different in the same prefecture (Figure 2).

### 2.2. Climatic Differences in Different Growth Seasons of Sugarcane

Differential analysis of the main climatic factors in three growth seasons of sugarcane showed significant differences in AAT, ARH, ARA, and ASD in different growth seasons (*p* < 0.01). AAT was the lowest in the late-growth season and highest in the mid-growth season. ARH was the lowest in the early-growth season and highest in the mid-growth season. ARA was the highest in the mid-growth season and lower in the late-growth season. ASD was the longest in the late-growth season and shortest in the mid-growth season (Figure 3).

### 2.3. Correlation between Climate Factors in Different Growth Periods and Sugarcane Production

#### 2.3.1. Correlation between AAT and Sugarcane Production

Linear analysis of the sugarcane yield and sucrose content of 26 sites from 2001/2002 to 2015/2016 with the corresponding AAT of early-growth, mid-growth, and late-growth seasons showed that AAT in the mid-growth season had a significant positive correlation with sucrose content (*p* < 0.05) and AAT of the late-growth season had a significant positive correlation with sucrose content (*p* < 0.01; Figure 4).

#### 2.3.2. Correlation between ARH and Sugarcane Production

Linear analysis of the sugarcane yield and sucrose content of 26 sites from 2001/2002 to 2015/2016 with the corresponding ARH of early-growth, mid-growth, and late-growth seasons showed that ARH in the mid-growth season had a significant positive correlation with sugarcane yield (*p* < 0.05; Figure 5).

#### 2.3.3. Correlation between ARA and Sugarcane Production

Linear analysis of the sugarcane yield and sucrose content of 26 sites from 2001/2002 to 2015/2016 with the corresponding ARA of early growth, mid-growth, and late-growth seasons showed that ARA in the early-growth season had a significant positive correlation with sugarcane yield (*p* < 0.05; Figure 6).

#### 2.3.4. Correlation between ASD and Sugarcane Production

Linear analysis of the sugarcane yield and sucrose content of 26 sites from 2001/2002 to 2015/2016 with the corresponding ASD of early-growth, mid-growth, and late-growth seasons showed that ASD in the late-growth season had a significant positive correlation with sugarcane yield (*p* < 0.05) and sucrose content (*p* < 0.01; Figure 7).

### 2.4. Division of Sugarcane Growth Climate Types

#### 2.4.1. Cluster Analysis of 26 Sites

According to the differences in AAT, ARH, ARA, and ASD in the early-growth, middle-growth, and late-growth seasons, 26 major sugarcane-producing regions were divided into five groups at a European distance of 70, where s7 was a separate group; s3 and s5 were the second group; s23, s10, s14, s16, s1, and s26 were the third group; and s21, s19, s25, s22, s6, and s20 were the fourth group. The rest of the sites formed the fifth group (Figure 8).

#### 2.4.2. Climate Differences among Different Cluster Groups

According to the differential analysis of climate factors of the five groups, AAT in the early-growth season of the second group was significantly lower than that of other groups and AAT in the early-growth season of the third group was significantly higher than that of other groups. ARA of the first group was significantly higher than that of other groups. The fifth group showed the longest ASD in the early-growth season, and the second group had the shortest ASD. The first group showed the highest ARH, and the fourth group had the lowest humidity in the early-growth season (Figure 9a). In the mid-growth season, the AAT of the first group was significantly lower than that of the other groups. The ARA of the first group was significantly higher than that of the other groups, and the ARA of the fourth group was significantly lower than that of the other groups. ASD values of the first and second groups were the lowest, and ASD values of the fourth and fifth groups were the highest. The ARH of the first group was the highest, and the ARH of the fourth group was the lowest (Figure 9b). In the late-growth season, the AAT of the first group was significantly lower than that of the other groups, and the ARA of the first group was significantly higher than that of the other groups, followed by the third group. The ATA values of the second and fourth groups were the lowest. The ASD of the fourth group was the longest and significantly higher than that of the other groups. The ARH was the highest in the first group and lowest in the fourth group (Figure 9c). The climate first type can be found in rainy and humid sugarcane areas characterized by high ARA and ARH during the entire growth period, low AAT and ASD in the mid-growth season, and low AAT in the late-growth season. The second type can be found in low-temperature and sunshine semi-humid sugarcane areas characterized by the lowest AAT in the early and middle stages of sugarcane growth, less ASD in the early and middle stages, and less ARA in the early and late stages. The third type can be found in high temperature and humidity sugarcane areas characterized by higher AAT and ARA, and moderate ASD during the entire growth period. The fourth type can be found in semi-humid and multi-sunshine sugarcane areas characterized by the lowest ARH in the entire growth period, lowest ARA in the middle and late seasons, and longest ASD. The fifth type can be found in humid and sunny areas characterized by the longest ASD and high ARH in the early and late seasons of sugarcane growth and moderate AAT and ARA during the entire growth season.

## 3. Discussion

Sugarcane is a crop that requires high temperature and humidity and strong light. Sugarcane planting is mainly concentrated between southern latitude 25° and northern latitude 25°. The growth of sugarcane has an important relationship with climate factors. Some researchers have reported that the growth of sugarcane is closely related to climate factors such as sunshine duration, temperature, rainfall, and humidity, and the factors are required in different amounts in different periods of sugarcane growth [16]. The climate characteristics of the low-latitude plateau of Yunnan are unique and cover the main climate types in China’s sugarcane regions, which are generally characterized by high temperature, humidity, and sunshine [17]. The low-latitude plateau of Yunnan is mountainous and hilly; sugarcane is mainly planted on dry and sloping land, which is barren and characterized by low soil moisture. Therefore, rainfall has become the main factor for the early growth of sugarcane in Yunnan. In this study, the climate of the sugarcane region in the low-latitude plateau of Yunnan was found to be generally dry in winter and early spring, with very little rainfall from November to March (0–50 mm; Figure 2). This season is also the key period for sugarcane planting. Therefore, early rainfall has become the main limiting factor for sugarcane emergence and tillering, which directly affects sugarcane yield. The accumulated rainfall of 10–20 days after sugarcane planting is the key meteorological factor that determines the emergence of sugarcane seedlings [18]. Water deficits can even reduce sugarcane yield by up to 60% [19], and the development of sugarcane is directed to areas with high water availability for growth and production [20]. In this study, ARA values in the mid-growth and late-growth seasons were not related to the sugarcane yield and sucrose content. Some studies have found that, in the field, sugarcane height, stem diameter, and yield are not easily affected by rainfall, which has little impact on sucrose content in the early harvest period [21]. Sugarcane ripening has been associated with both incident sunlight and temperature, but not with rainfall [11], and sugarcane production was reduced in only the dry and rainy years [22]. This is consistent with the results of this study; we found that rainfall in the mid-growth season was sufficient and rainfall in the later-growth season was low (Figure 3c), which are favorable conditions for sugarcane growth on the basis of the analysis of rainfall in the last 20 years.

Temperature is the main factor that affects the germination of sugarcane buds. Within a certain range, with an increase in temperature, the activity of enzymes in the seedlings increases as respiratory metabolism increases, and the germination speed of seed buds accelerates. The minimum temperature for bud germination in most varieties is 13 °C, and the optimum temperature is 20–30 °C; the minimum temperature for rooting is 10 °C, and the optimum temperature is 25 °C [8]. The analysis of 26 major sugarcane-producing areas showed that the lowest AAT in the low-latitude plateau area was mainly in January and February (10–13 °C), and the temperature began to increase in March (at about 15 °C) and then increased month by month. The highest average temperature was in June and July (about 25 °C), and the temperature characteristics in this area were more suitable for sugarcane growth. We found that the AAT in the middle and late stages affected the sucrose content of sugarcane and detected a significant positive correlation between temperature and sucrose content. A number of other modelling studies have reported the positive effect of increased temperatures on sugarcane yield [23]. Lower maximum and higher minimum temperatures and light wind have been found to be favorable for the growth of sugarcane in the active and elongation stages [24]. Luo et al. reported that temperature has a lower impact on sugarcane growth; temperature mainly affects sucrose content [21], but it has little impact on other aspects. The emergence rate, plant emergence rate, and tillering rate of sugarcane are significantly affected by rainfall, whereas plant height, stem diameter, and yield are not easily affected by rainfall. Cold and freezing injury of sugarcane is one of the main meteorological disasters in the low-latitude plateau region of Yunnan [25]. The low-temperature and high-humidity conditions during the start of the crushing season affect sugar accumulation in sugarcane [23]. In this study, sugarcane growth was divided into three periods. In the mid-growth season, the rainfall was sufficient and temperature and radiation became the main limiting factors for sugarcane growth and sucrose accumulation. Since 2008, severe low temperatures and drought were recorded almost every year, which caused a continuous decrease in cane and sucrose productivity in Guangxi and Yunnan; the decreasing trend reached the bottom in the milling year 2010/2011 [26]. In the late-growth season, low temperature is the main threat to sugarcane production, and an increase in average accumulated temperature is beneficial to the increase in sucrose content in sugarcane.

Rupa found that maximum and minimum temperatures and relative humidity during the first three months of crop production (germination and tillering phases) have a profound influence on the yield [27]. In this study, a significant correlation was found between relative humidity and sugarcane yield in the mid-growth season, which is consistent with the results of a previous study. The yields of sugarcane are strongly correlated across irrigated and rainfed environments [28]. A previous study revealed that maximum temperature, morning humidity, and sunshine hours play the most important roles during the germination stage Generally, variations in relative humidity result in flowering in certain varieties of sugarcane [29] and occurrence of the sugarcane leaf disease [30,31]. In the low-latitude plateau area, ASD was the longest in the late-growth season of sugarcane and shortest in the mid-growth season. In this study, ASD was found to be average (>200 h/month) in the late-growth stage, with a significant positive correlation between the yield and sucrose content of sugarcane. Sugarcane plant height and stem diameter have been significantly or extremely significantly correlated with average temperature, precipitation, and sunshine duration [18]. Huang also found that, in the early stage of sugarcane growth (from April to June) [32], the three factors of sunshine duration, temperature, and water were relatively well-matched and the seedling condition was good, which laid a good foundation for the later high yield of sugarcane. In sugarcane, sucrose accumulation and ASD showed a significant positive correlation (*p* < 0.01) [33]. In fact, the yield and sucrose content of sugarcane are not affected by a single climatic factor, but by multiple factors. The growth and yield of sugarcane are profoundly influenced by climate elements. This may be due to a large growing season for good phenological development, high germination, more tillers, and maximally millable canes. Appropriate temperature, sunshine duration, humidity, rainfall, etc., are favorable for germination, tillering, and cane formation.

Sugarcane planting can be subdivided into four growth stages according to physiological and growth characteristics: seed germination, tillering, stalk elongation to maturity, and flowering. In this study, sugarcane planting was divided into three important growth periods according to seasonal changes: early-growth season, mid-growth season, and late-growth season. Meteorological factors such as AAT, ARH, ARA, and ASD were found to significantly affect the growth of sugarcane in different seasons (*p* < 0.05, *p* < 0.01; Figure 4, Figure 5, Figure 6 and Figure 7). Moreover, in different growth seasons, these meteorological factors have different effects on sugarcane yield and sucrose content. Therefore, according to the differences in meteorological factors in different growth seasons, the low-latitude plateau of Yunnan was divided into five different ecological types (Figure 8). This will help us understand the growth potential of different ecological types of sugarcane, which is of great significance in improving productivity. To identify areas suitable for expansion, the Brazilian government performed Sugarcane Agroecological Zoning (AEZ-Sugarcane); Silva et al. modeled sugarcane expansion in São Paulo in 2041 and 2060 to map low-risk areas on the basis of climate change scenarios, in addition to comparing the expansion areas from the model with suitable AEZ-Sugarcane areas [34]. According to the ecological adaptability of sugarcane, 69 sugarcane production areas in Guangxi, Guangdong, and Yunnan in China were divided into six groups of ecological sugarcane areas on the basis of natural climate characteristics of the main sugarcane production areas, namely, subtropical humid low-light sugarcane areas, subtropical humid low-evaporation sugarcane areas, tropical humid sugarcane areas, subtropical semi-humid sugarcane areas, subtropical high-humidity sugarcane areas, and subtropical humid multi-light sugarcane areas [17]. The climatic types of the main sugarcane-producing areas in Yunnan were classified using the systematic clustering method. The six sugarcane climate types have good regional distribution; the climate characteristics of each type have obvious differences, which are mainly reflected in the average sunshine hours from April to October, average precipitation from May to October, average sunshine hours from November to February, and accumulated temperature differences from September to December [35]. The division of climate types in the sugarcane-growing areas provided a reference for our research. However, it is essentially different from previous studies; the division of climate types is more detailed and a significant guide for production. In this study, 26 major sugarcane-producing counties in the low-latitude plateau region could be divided into five climatic ecological types for sugarcane growth. In the first ecological type, both ARA and ARH are high throughout the growth period, AAT and ASD are low in the mid-growth season, and AAT is low in the late-growth season of sugarcane. Using Jinping County (s2) as the representative model, this type is conducive to an increase in sugarcane yield and not conducive to an increase in sucrose content. In the second ecological type, AAT is low in the early and middle seasons of sugarcane growth, the ASD is short in the early season, and ARA is low in the late season, with Funing (s3) and Guangnan County (s5) as the representative models. This type of sugarcane has a medium yield, which is unfavorable for an increase in sucrose content. In the third ecological type, the AAT of sugarcane is significantly higher than that of other sugarcane areas in the early-growth season, rainfall and humidity are relatively high in the late-growth season, and climatic characteristics in the mid-growth season are not obvious, with Cangyuan (s1), Lancang (s10), Mangshi (s14), Mengla (s16), Ximeng (s23), and Zhenkang County (s26) as the representative models. This type of sugarcane has a moderate yield and sucrose content, which is not particularly conducive or detrimental to an increase in sugarcane and sucrose yield. In the fourth ecological type, air humidity is the lowest in the early, middle, and late seasons of sugarcane growth, rainfall is the lowest in the middle and late seasons, and light duration is the longest, with Honghe (s6), Shidian (s19), Shiping (s20), Shuangjiang (s21), Wenshan (s22), and Yunxian (s25) as the representative models. This type is conducive to an increase in sucrose content, and the yield is medium. In the fifth ecological type, light duration is the longest in the early and late seasons of sugarcane growth and rainfall is relatively low, with Fengqing (s2), Menghai (s15), Linxiang District (s12), Lianghe (s11), Yingjiang (s24), Jingdong (s8), Gengma (s4), Longchuan (s13), Ruili (s18), Jinggu (s9), and Menglian County (s17) as the representative models. This type is beneficial to sugarcane yield and sucrose content. Riajaya found that the rainy season in sugarcane regions in Indonesia ranges from 20–25 ten-day periods in 59% of the seasonal zone in Sumatra [20], 15–20 ten-day periods in 73% of the regions in Java and Madura, and 20–25 and 10–15 ten-day periods in 33% and 34% of the regions in Sulawesi. Areas with 25–30 ten-day rainy periods are more suitable for the development of sugarcane as the raw material for the brown sucrose industry, and areas with 10–15 ten-day rainy periods require additional irrigation to meet the water needs of sugarcane plants. Areas with 15–20 ten-day rainy periods are ideal for sugarcane growth. In this study, the climate types were precisely divided, which will help guide sugarcane production in the low-latitude plateau region.

## 4. Materials and Methods

### 4.1. Data Collection and Test Sites

#### 4.1.1. Data Collection

The data collected in this study were mainly divided into two categories obtained from the sugarcane planting (test) region. The above-mentioned data were provided by the archives of the Climate Center of Yunnan Meteorological Bureau. The second category was composed of data of the annual sugarcane-planting area, sugarcane harvest amount, and average sucrose content of sugarcane in each major sugarcane-producing county (test sites) from 2001/2002 to 2018/2019. The average yield of sugarcane was calculated from the sugarcane harvest amount divided by the sugarcane planting area. The above-mentioned data were provided by the sugarcane research institute of the Yunnan Academy of Agricultural Sciences. The yield and sucrose content were the average values of New plant, Ratoon1, and Ratoon2 planting periods collected from varieties Roc22 and YT93-159. The test soil type was clay; organic matter content, 12.5–18.6 g/kg; pH value, 5.1–5.9; available phosphorus, 32.3–52.1 mg/kg; available potassium, 42.1–56.0 mg/kg; and alkali hydrolyzed nitrogen, 57.9–72.3 mg/kg. Roc22 (planting areas, 60%) and YT93-159 (planting areas, 40%) were planted at an altitude between 826 m and 1100 m by using one-time fertilization and full-film coverage technology with a row spacing of 1.1 m. The planting time of each variety was between January and February.

#### 4.1.2. Overview of Test Sites

In this study, the data of climatic factors, yield, and sucrose content of sugarcane from 26 counties (test sites) in seven prefecture-level cities that represented the main sugarcane-producing areas were collected from the low-latitude plateau of Yunnan (21° N–25° N, 97° E–106° E) between 2000 and 2019. The locations of the test sites were as follows: Lianghe County (s11), Longchuan County (s13), Ruili City (s18), Mangshi (s14) and Yingjiang County (s24) in Dehong prefecture-level city, Cangyuan County (s1), Fengqing County (s2), Gengma County (s4), Linxiang District (s12), Shuangjiang County (s21), Zhenkang County (s26) and Yun County (s25) in Lincang prefecture-level city, Shidian County (s19) in Baoshan prefecture-level city, Shiping County (s20), Jinping County (s7) and Honghe County (s6) in Honghe prefecture-level city, Jingdong County (s8), Jinggu County (s9), Lancang County (s10), Menglian County (s17) and Ximeng County (s23) in Pu’er prefecture-level city, Menghai County (s15) and Mengla County (s16) in Xishuangbanna prefecture-level city, Funing County (s3), and Guangnan County (s5) and Wenshan City (s22) in Wenshan prefecture-level city. They represent more than 85% of the sugarcane-planting area in Yunnan, with an altitude of 500 to 2000 m. The annual AAT in the last 20 years was mainly between 19.0 and 20.1 °C; annual ARH, between 73% and 79%; annual ARA, between 50 mm and 140 mm; and annual ASD, between 1800 and 2400 h (Figure 1).

### 4.2. Study Factors

#### 4.2.1. Division of Sugarcane Growth Season

The sugarcane growth season was divided into three periods, namely, early-growth season (seedling stage and tillering stage), mid-growth season (jointing stage and big growth stage), and late-growth season (formation and stability stage for sugarcane yield and sucrose content). On the basis of the planting habit and growth characteristics of spring sugarcane in the low-latitude plateau of Yunnan, the early-growth season was from March to June, the mid-growth season was from July to October, and the late-growth season was from November to February.

#### 4.2.2. Climatic Characteristics of Growth Seasons and Sugarcane Production

The average sugarcane yield and sucrose content data were collected from 26 sites between 2005/2006 and 2015/2016, and the data for AAT, ARH, ARA, and ASD in the corresponding years were also obtained. Correlation analysis of the main climatic factors in different growth seasons with sugarcane yield and sucrose content was performed to study the relationship between climate and sugarcane production. A summary of average annual AAT, ARH, AAR, AAS, yield, and sucrose content of the corresponding sites and years has been provided in Appendix A.

#### 4.2.3. Classification of Climate Types on the Basis of Sugarcane Growth

According to the climate variation characteristics of the 26 sites in different growth seasons, the climate types of the sugarcane-planting region in the low-latitude plateau of Yunnan were classified and differences among the different climate types were analyzed.

### 4.3. Statistical Analysis

Microsoft Excel 2019 (Microsoft, Redmond, WA, USA) was used for sorting the data of the climate, sugarcane yield, and sucrose content. DPS v14.10 software (Zhejiang University, Hangzhou, China) was used for the statistical analysis, including the calculation of mean value, standard deviation, and coefficient of variation. R software (version 4.1.0, R core team, 2021) was used for creating the figures. The package “ggplot2” of R software was used to create the line charts, box diagrams, scatter charts, and fitting curve and perform the cluster analysis. The package “ggsignif” was used for the LSD test (Detailed programming codes can be obtained from us).

## 5. Conclusions

The climate in different seasons in the low-latitude plateau region was significantly different, which has varied effects on sugarcane yield and sucrose content. Winter and early spring (from November to March) are dry and rainless, which is not conducive to the germination and growth of sugarcane buds. The rainfall is mainly concentrated in June to September, which is conducive to an increase in sugarcane yield. The low-temperature period is short, and the lowest temperature is mainly concentrated in December and January. The difference in sunshine duration in the main sugarcane-producing region was significant, which was the main factor that affected the sugarcane yield and sucrose content. The relative humidity of sugarcane was relatively high throughout the growth season. Average air temperature in the mid-growth season and late-growth season significantly affected sucrose content, and low temperature was not conducive to sucrose formation. Average relative humidity in the mid-growth season affected sugarcane yield, and high humidity was beneficial to an increase in yield. Average rainfall amount in the early-growth season affected the sugarcane yield, mainly affecting the emergence and tillering of sugarcane. Average sunshine duration in the late-growth season was the main factor that affected the yield and sucrose content of sugarcane, and it was beneficial for high yield and sucrose content. The low-latitude plateau region has five climate types that contribute to different sugarcane yields and sucrose contents. The rainy and humid type contribute to a high sugarcane yield, but not to a high sucrose content. The low temperature and sunshine semi-humid type is unfavorable for a high sugarcane yield and sucrose content. The high temperature and humidity type has good sugarcane growth potential and contributes to sugarcane yield and sucrose content. The semi-humid and multi-sunshine type contributes to an increase in sucrose content. The humid and sunny type is beneficial for a high sugarcane yield and sucrose content.

## Figures and Tables

**Figure 1 plants-12-02712-f001:**
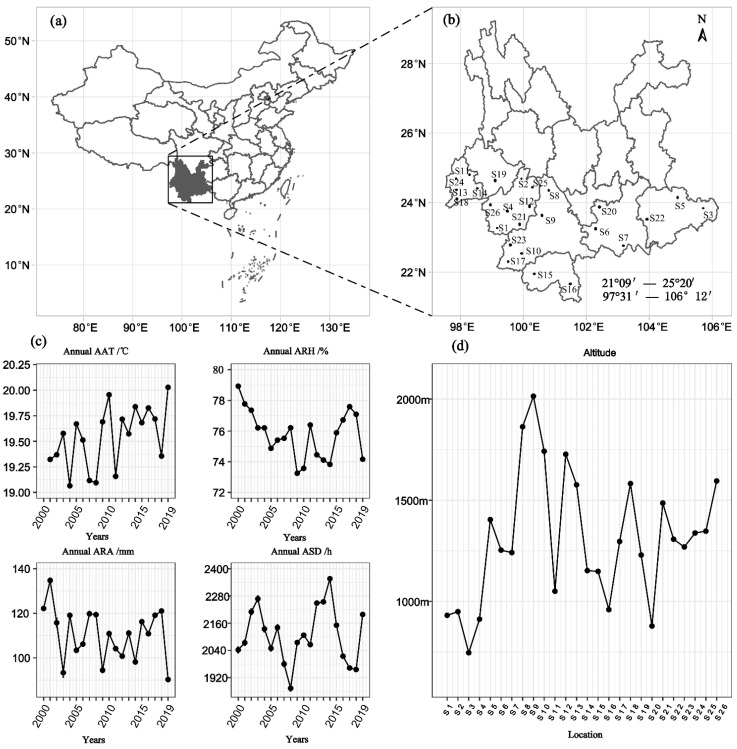
Study area. (**a**) Location of Yunnan in China. (**b**) Distribution of the study sites. (**c**) Annual average air temperature (AAT), average relative humidity (ARH), average rainfall amount (ARA), and average sunshine duration (ASD) across all sites over 20 years. (**d**) Average altitude of the test sites.

**Figure 2 plants-12-02712-f002:**
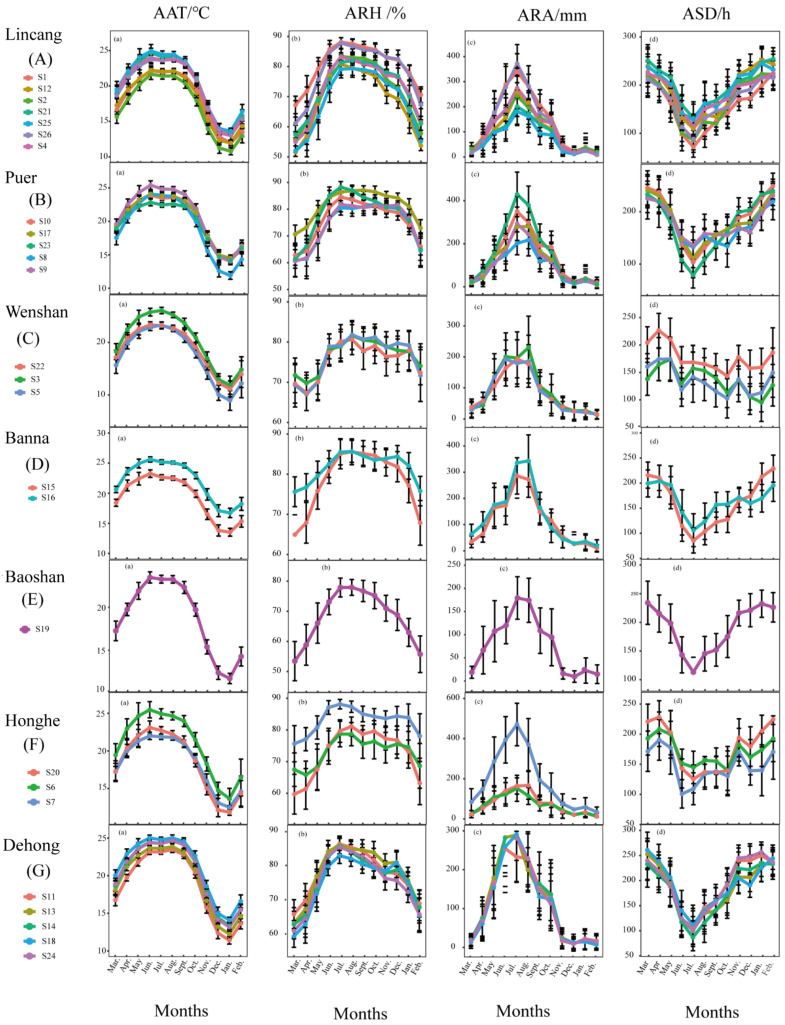
(a) Monthly average air temperature (AAT), (b) average relative humidity (ARH), (c) average rainfall amount (ARA), and (d) average sunshine duration (ASD) in 20 years (2000 to 2019) in 26 counties in seven prefectures in the low-latitude plateau of Yunnan. Error bars show standard error of the mean.

**Figure 3 plants-12-02712-f003:**
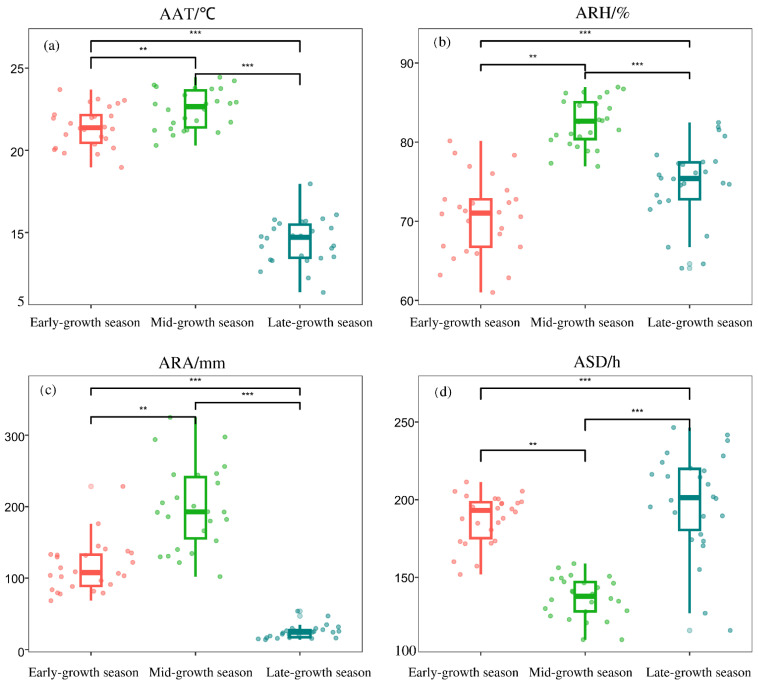
Differential analysis of average air temperature (AAT), average relative humidity (ARH), average rainfall amount (ARA), and average sunshine duration (ASD) in different sugarcane growth seasons. (**a**) average air temperature (AAT) in different seasons. (**b**) average relative humidity (ARH) in different seasons. (**c**) average rainfall amount (ARA) in different seasons. (**d**) average sunshine duration (ASD) in different seasons. ** and *** *F*-values are significant at *p* < 0.01 and *p* < 0.001 levels, respectively. Scatter plots show the mean of weather data from 2000 to 2019 at 26 sites.

**Figure 4 plants-12-02712-f004:**
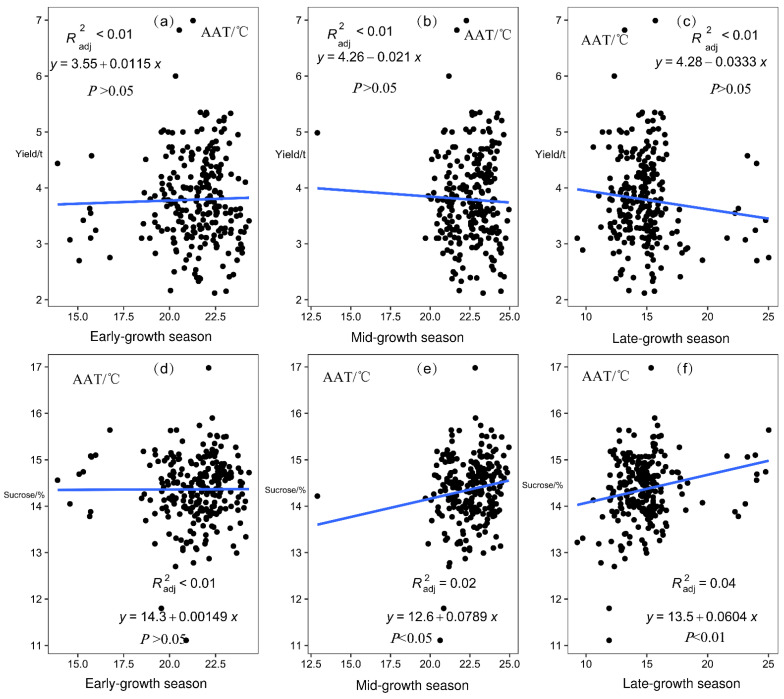
Linear analysis of average air temperature (AAT) and yield and sucrose content across all sites in different sugarcane growth seasons. (**a**) Linear analysis of average air temperature (AAT) and yield in early-growth season. (**b**) Linear analysis of average air temperature (AAT) and yield in mid-growth season. (**c**) Linear analysis of average air temperature (AAT) and yield in late-growth season. (**d**) Linear analysis of average air temperature (AAT) and sucrose in early-growth season. (**e**) Linear analysis of average air temperature (AAT) and sucrose in mid-growth season. (**f**) Linear analysis of average air temperature (AAT) and sucrose in late-growth season. Scatter plots show the mean of weather data from 2005/2006 to 2015/2016 at 26 sites.

**Figure 5 plants-12-02712-f005:**
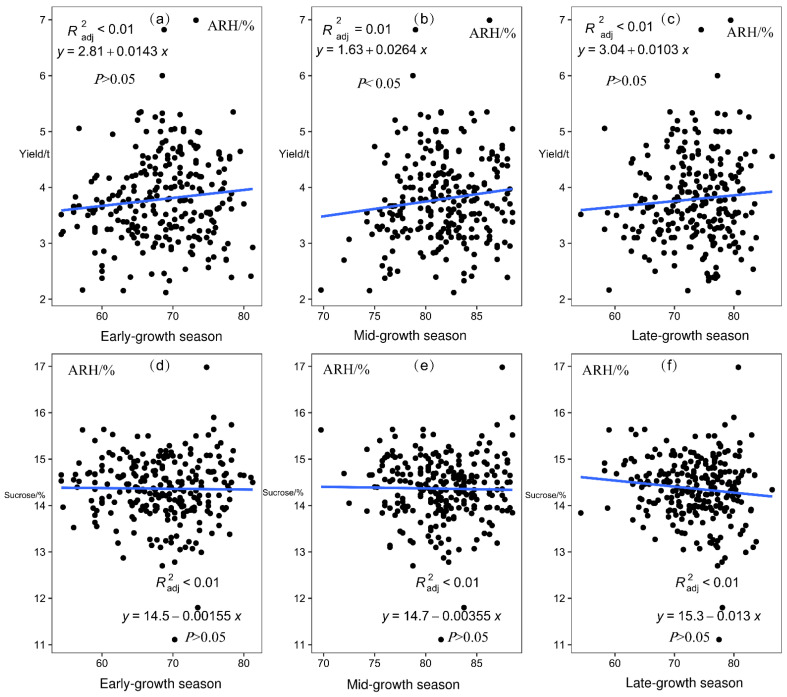
Linear analysis of average relative humidity (ARH) and yield and sucrose content across all sites in different sugarcane growth seasons. (**a**) Linear analysis of average relative humidity (ARH) and yield in early-growth season. (**b**) Linear analysis of average relative humidity (ARH) and yield in mid-growth season. (**c**) Linear analysis of average relative humidity (ARH) and yield in late-growth season. (**d**) Linear analysis of average relative humidity (ARH) and sucrose in early-growth season. (**e**) Linear analysis of average relative humidity (ARH) and sucrose in mid-growth season. (**f**) Linear analysis of average relative humidity (ARH) and sucrose in late-growth season. Scatter plots show the mean of weather data from 2005/2006 to 2015/2016 at 26 sites.

**Figure 6 plants-12-02712-f006:**
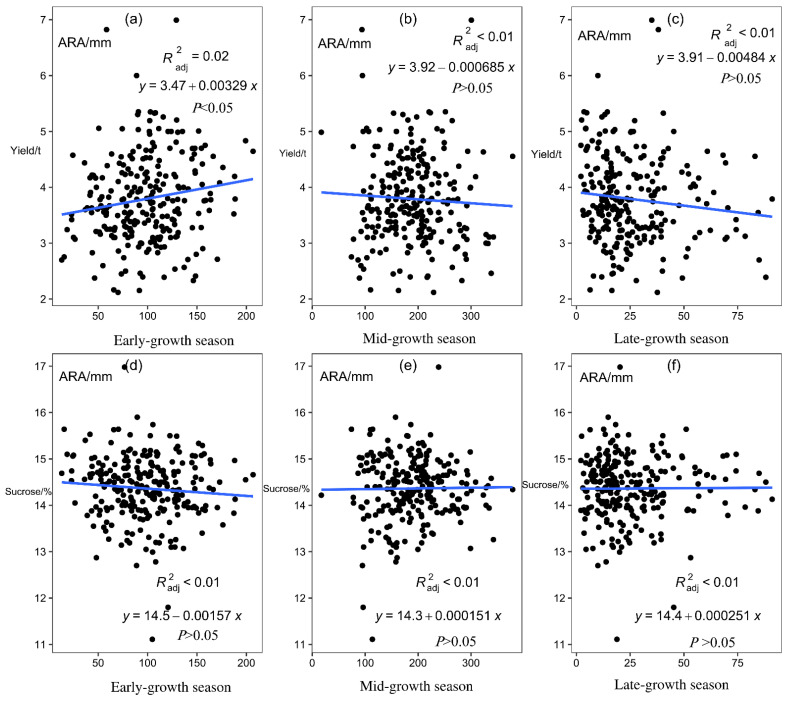
Linear analysis of average rainfall amount (ARA) and yield and sucrose content across all sites in different sugarcane growth seasons. (**a**) Linear analysis of average rainfall amount (ARA) and yield in early-growth season. (**b**) Linear analysis of average rainfall amount (ARA) and yield in mid-growth season. (**c**) Linear analysis of average rainfall amount (ARA) and yield in late-growth season. (**d**) Linear analysis of average rainfall amount (ARA) and sucrose in early-growth season. (**e**) Linear analysis of average rainfall amount (ARA) and sucrose in mid-growth season. (**f**) Linear analysis of average rainfall amount (ARA) and sucrose in late-growth season. Scatter plots show the mean of weather data from 2005/2006 to 2015/2016 at 26 sites.

**Figure 7 plants-12-02712-f007:**
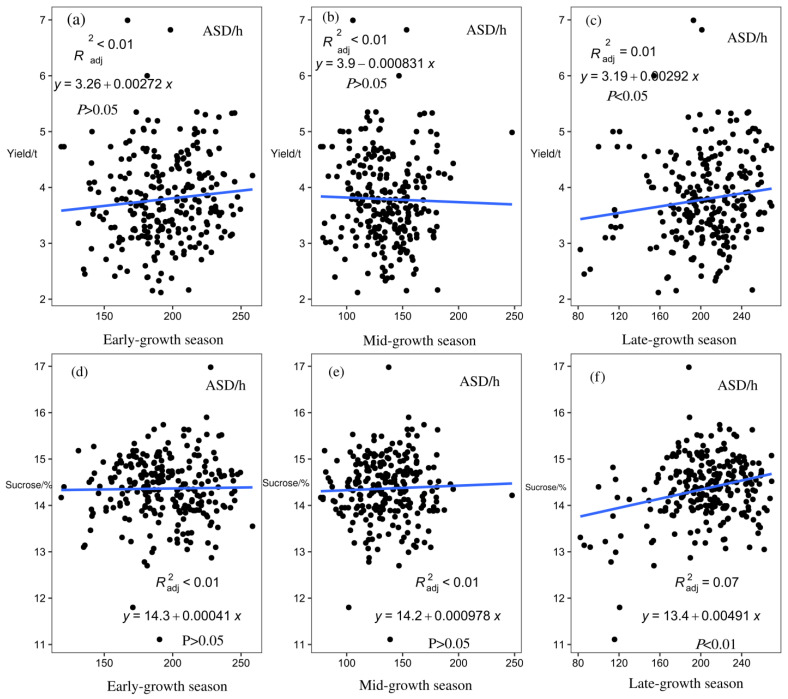
Linear analysis of average sunshine duration (ASD) and yield and sucrose content across all sites in different sugarcane growth seasons. (**a**) Linear analysis of average sunshine duration (ASD) and yield in early-growth season. (**b**) Linear analysis of average sunshine duration (ASD) and yield in mid-growth season. (**c**) Linear analysis of average sunshine duration (ASD) and yield in late-growth season. (**d**) Linear analysis of average sunshine duration (ASD) and sucrose in early-growth season. (**e**) Linear analysis of average sunshine duration (ASD) and sucrose in mid-growth season. (**f**) Linear analysis of average sunshine duration (ASD) and sucrose in late-growth season. Scatter plots show the mean of weather data from 2005/2006 to 2015/2016 at 26 sites.

**Figure 8 plants-12-02712-f008:**
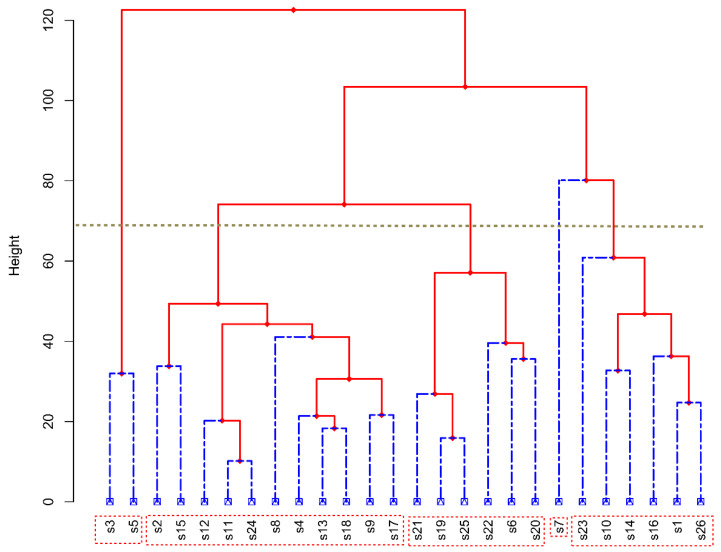
Cluster analysis of 26 sites on the basis of average air temperature (AAT), average relative humidity (ARH), average rainfall amount (ARA), and average sunshine duration (ASD) from 2000 to 2019 (20 years) in different sugarcane growth seasons.

**Figure 9 plants-12-02712-f009:**
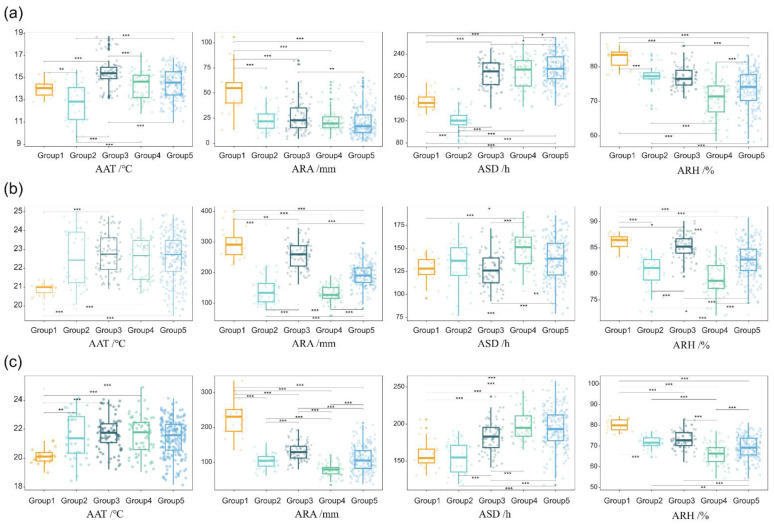
Differences in the average air temperature (AAT), average relative humidity (ARH), average rainfall amount (ARA), and average sunshine duration (ASD) of five cluster groups in three growth seasons. *, **, and *** *F*-values significant at *p* < 0.05, *p* < 0.01, and *p* < 0.001 levels, respectively. (**a**) Early-growth season; (**b**) mid-growth season; (**c**) later-growth season.

## Data Availability

Raw data are available upon request.

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
