# Peer review of "Climate Variations in the Low-Latitude Plateau Contribute to Different Sugarcane (Saccharum spp.) Yields and Sugar Contents in China"

_plants, 2023, doi:10.3390/plants12142712_

Round 1
Reviewer 1 Report
Thank you for the opportunity to review this manuscript titled, “Climate variations in the low-latitude plateau contribute to different sugarcane (Saccharum spp.) yields and sugar contents in China”. This approach is a creative way to bring a systematized and quantitative framing to understand this now large body of literature on climate variability. However, I have some concerns that author should consider to improve the manuscript.
- The abstract should state briefly the purpose of the research, the principal results and major conclusions.
- The introduction section should be reorganized, provide additional information related to the methods followed in previous studies and highlight the gap of knowledge that this study seeks to fill.
- Could the authors provide more information on the control the validity of the results?
- Much more explanations and interpretations must be added for the Results, which are not enough.
- Please make sure your conclusions' section underscore the scientific value added of your paper, and/or the applicability of your findings/results, as indicated previously.
Minor editing of English language required
Reviewer 2 Report
The manuscript is well written. By the way minor revision will further improve its quality, Check English. Use journal format for citation (numbering), in some places authors names given with no proper numbering. Do not start sentence with numerical values (eg., 26). Also write text for numbers less than 10 (eg,, 3 be replaced with three, etc.). conclusions must be based on results obtained, no need of general comments. Also check the attached file, please.

minor revision is needed, just check again for errors
Reviewer 3 Report
General Comments
The present article focuses on an important issue within the scope of the Journal. It provides valuable and valid data, results, and conclusions. However, a substantial revision is necessary to improve objectivity and clarity. The manuscript is currently burdened with excessive length and redundant information. It is recommended to reduce the number of figures and subsections and eliminate repetitions and unnecessary information.
Specific Comments
1. The title needs to be revised as it reads more like a conclusion than a concise title. It is also preferable to avoid the simultaneous use of the common name and the scientific name of sugarcane in the title, reinforced by the fact that they are already included as keywords.
2. The abstract is excessively lengthy and should be condensed by half. A more objective approach is recommended, focusing solely on the main results of the research and avoiding excessive detail.
3. Line 13, The use of the first person should be avoided, correct throughout the entire manuscript
4. Line 42, Remove keywords abbreviations.
5. The Results section must be positioned after the Materials and Methods section. Materials and Methods must be the section 2.
6. Lines 498 and 510, "from 2019 to 2000" instead of “from 2020 to 2019”.
7. The information presented in sub-section 4.2.1 was already mentioned in section 4.1, therefore, sub-section 4.2.1 can be omitted. Similarly, sub-section 4.2.4 can be also omitted, as it does not provide relevant information. However, if the authors believe it is necessary, they can incorporate this information into a previous sub-section.
8. Tables 1 and 2 should be removed from the main manuscript and placed as supplementary material.
9. The quality of Figure 1 needs to be improved. It is not possible to read the labels of the Y axes of Figure 1c) and 1 d).
10. Sessions 2.1.1 to sections 2.1.7 should be condensed into a single section, significantly reducing the text. Additionally, the figures of these sections should be removed and placed in the supplementary material.
Minor editing of English language required
Round 2
Reviewer 1 Report
Accept for publication.
Minor editing of English language required
Reviewer 3 Report
The authors answered my questions and made the suggested changes, demonstrating an excellent work rearranging the figures. However, the summary is still a bit too long, but that's acceptable.
Line 764, Table 1 and Table 2 should be cited as supplementary material, namely as “Table S1 and Table S2”, and the same correction should be made in the respective Tables’ titles.
Minor editing of English language required
